# Health Promotion Programs Can Mitigate Public Spending on Hospitalizations for Stroke: An Econometric Analysis of the Health Gym Program in the State of Pernambuco, Brazil

**DOI:** 10.3390/ijerph191912174

**Published:** 2022-09-26

**Authors:** Flávio Renato Barros da Guarda

**Affiliations:** Department of Public Heath, Federal University of Pernambuco, Recife 50670-901, Brazil; flaviodaguarda@hotmail.com; Tel.: +55-81-3114-4101

**Keywords:** hospitalization, propensity score, difference-in-differences, unified health system, motor activity, impact evaluation, stroke, policy analysis, primary health care, healthcare costs

## Abstract

Health promotion programs can reduce morbidity and mortality from chronic diseases, as well as public spending on health. The current study aims to evaluate the effects of the Health Gym Program on expenditures on hospitalizations for stroke in the state of Pernambuco, Brazil. This public policy impact assessment used an econometric model that combines the difference-in-difference estimator with propensity score matching. Data referring to the health, demographic, and socioeconomic characteristics of the 185 municipalities in Pernambuco were collected for the period from 2007 to 2019. Validation tests were carried out of the empirical strategy, the estimation of models with fixed effects for multiple periods and validation post-tests, and robustness of the results. In total, US$ 52,141,798.71 was spent on hospitalizations for stroke, corresponding to 4.42% of the expenses on hospitalizations for all causes over the period studied. Municipalities that implemented the Health Gym Program spent 17.85% less on hospitalizations for stroke than municipalities that did not. The findings of this study indicate that the Health Gym Program was effective in reducing expenses with hospitalizations for stroke and that its implementation has the potential to reduce expenses related to rehabilitation, sick leave, and early retirement.

## 1. Introduction

Cerebrovascular diseases are an important public health problem and occupy the second position among the deadliest diseases in the world [1]. These diseases include the Cerebral Vascular Accident (stroke), which is considered one of the main global causes of illness [2,3] and was the primary cause of hospital admissions in the subgroup of diseases of the circulatory system in Brazil in 2018, with 69.84 hospitalizations for each group of 100,000 inhabitants [4].

The occurrence of hospitalizations for this disease in the Brazilian population is more frequent among men, in individuals with less schooling, and in people over 65 years of age [5]. In addition, morbidity and mortality after stroke are associated with characteristics particular to the municipalities, such as the Human Development Index [6] and the Gross Domestic Product [7].

In addition to compromising the health and quality of life of affected individuals, the occurrence of stroke demands significant expenditure on health actions and services, especially hospitalizations. Expenditure on financial transfers from the federal government for the cost of hospital admissions in the municipalities was US$ 49,806,982.90, which represented 1.3% of the expenditure on hospitalizations for all causes in 2018 [4]. In addition, the average expenditure on the rehabilitation of stroke victims in the public health system in the second most populous state in Brazil was US$ 305.18 [8], which represents 89% of the total expenditure on stroke treatment [5,8].

The main risk factors for stroke are preventable or controllable (obesity, dyslipidemia, smoking, hypertension, and physical inactivity) [9,10]. In this sense, health promotion and primary prevention actions should focus on primary health care, aiming to reduce exposure to risk factors, the occurrence of new events, and worsening of the patient’s clinical condition [11].

Health promotion actions can reduce the incidence and costs of hospital admissions and rehabilitation of patients who have suffered a stroke [12,13]. Lima et al. (2020) [14] evaluated the impact of a health promotion program on expenses with hospital admissions for cerebrovascular diseases and observed that municipalities which implemented the intervention spent, on average, US$ 1258.61 less for each group of 10 thousand inhabitants, when compared to municipalities without the program. This value represents an annual savings of US$ 1,069,818.50 with hospital admissions.

The Brazilian Ministry of Health has invested in health promotion policies and programs [15,16], among which the Health Gym Program (HGP) stands out. This program uses financial resources from the federal government for municipalities to build and fund public spaces, intending to expand the scope of health promotion actions within the range of primary health care [15,17].

The HGP has been implemented in more than 1500 Brazilian cities and its guidelines indicate that it should be a reference space for health promotion, care production, and prevention and control of non-communicable chronic diseases in municipalities [15,18]. The program is still cited as the main health promotion strategy in the text of the national primary care policy and in the Plan to Combat Chronic Diseases proposed by the Ministry of Health for the period from 2011 to 2022 [15].

The state of Pernambuco has 267 centers of the HGP in 134 municipalities [19]. These were implemented in 2011, mainly through the incorporation into other similar health promotion programs already established in the municipalities, which enabled Pernambuco to initiate HGP actions even before the other states of the federation began to build their centers [15].

Evidence points out that health promotion programs can mitigate the occurrence of cases and expenses with hospitalizations for stroke [13,14], and that the HGP was effective in increasing the level of physical activity of the population [20] and reducing the mortality from arterial hypertension [21]. However, the relationship between the implementation of the program and public spending on health is still unclear. In this sense, it is necessary to evaluate the impact of this intervention, to measure its effectiveness, generate evidence capable of justifying the opportunity cost related to public investment in this intervention, and support decision-making processes regarding its maintenance or expansion [22]. In this sense, the objective of the current study is to evaluate the effects of the HGP on hospital admission expenses for stroke in the state of Pernambuco from 2011 to 2019.

## 2. Materials and Methods

### 2.1. Study Design and Empirical Strategy

This study is characterized as an impact assessment of public policies, which aims to measure the effects of the HGP on hospital admission expenses for stroke in the municipalities that implemented this intervention, compared to those that did not. The empirical strategy makes use of econometric models that combine the difference-in-difference estimator (DID) with Propensity Score Matching (PSM).

The difference-in-difference estimator is a method widely used in quasi-experimental approaches to evaluate the impact of policies [23,24]. It is used to calculate the difference of the differences in the results observed in the groups of treated units and controls in the periods before and after the implementation of the policy [23,24,25]. The DID reduces the risk of bias related to unobservable characteristics of the municipalities, which may affect the outcome variable (expenditure on hospitalizations for stroke) [23,24,25,26].

PSM identifies (and matches) untreated units that are similar to treated units in their observable characteristics and compares them against the mean values of the outcome variable. In this sense, PSM minimizes not only the biases arising from the distribution of observable characteristics, but also those related to the absence of common support [27,28].

### 2.2. Databases and Study Variables

The database is annual for the period from 2007 (four years before the implementation of the HGP) to 2019 (eight years after its implementation) and consists of all 185 municipalities in Pernambuco, constituting 2207 observations. The 134 municipalities that implemented the HGP as of 2011 are considered as treated and are the focus of the analysis. The remaining 51 municipalities are the counterfactuals of the treated and were designated as controls.

The outcome variable for this study is the natural logarithm of expenditure on hospital admissions for stroke (in individuals of both sexes and aged 40 years or older), according to the patient’s place of residence, and considering the Hospitalization Authorizations paid and registered in the National Hospital Information System (SIH-DATASUS). Considering that expenditure on hospitalizations had a value of zero for some observations, the strategy of adding one unit to the original amount of expenditure before converting to the natural logarithm was adopted, as recommended by Wooldridge (2016) [29].

The control variables were selected from epidemiological models that detail the health, demographic, and socioeconomic aspects associated with hospital admissions for circulatory system diseases, cerebrovascular diseases, and stroke in the Brazilian population [30,31,32].

The data referring to the health characteristics in the municipalities were: the natural logarithm of the number of doctors in each municipality; the number of hospital beds in the public health network; and the presence of Multiprofessional Support Teams for primary health care actions (NASF-AB), all of which were obtained from the National Registry of Health Establishments (CNES/DATASUS).

For the demographic variables, we considered the rate of individuals over 40 years of age per 10,000 inhabitants (calculated using data from the Brazilian Institute of Geography and Statistics—IBGE) and the pass rate in high school, which was obtained from the website of the National Institute of Educational Studies and Research (INEP).

The socioeconomic variables of the municipalities were the Gross Domestic Product per capita (collected from the website of the Brazilian Institute of Geography and Statistics), and the total health expenditure, collected from the Information System on Public Health Budgets (SIOPS).

All variables were collected from national official data, which comprise the health information systems of the Brazilian Ministry of Health, data from the Brazilian population census, official statistics from the Ministry of Education, and the national public budget system. Data on expenditures on hospital admissions for stroke represent amounts that are transferred by the ministry of health to municipalities, undergo internal audits, and follow national protocols for collection, recording, analysis, and dissemination.

Some municipalities presented values above the average of the other municipalities for the majority of variables and were considered as outliers. These values can distort the matching, compromise DID estimates, and lead to type I and II errors, as they may alter the metrics for defining good counterfactuals, and/or violate the balancing criterion used to specify the propensity score [33]. In this way, a binary variable indicating whether a municipality was an outlier was included in the model [34], aiming to identify differentiated patterns (both superior and inferior) in the indicators of the municipalities, resulting in better distribution of the data.

### 2.3. Data Analysis

All data analyses in this study were performed in STATA^®^ software version 16.0 and the results are presented in tables.

The characterization of the municipalities and the expenses with hospital admissions for stroke was performed using descriptive statistics procedures (frequencies, means, and standard deviations). To verify the differences between means, the Student’s *t*-test was used. The evaluation of the effects of the HGP on hospitalization expenses for stroke was performed using a PSM-DID estimation strategy in a Fixed Effect data model for multiple periods. The analytical procedures involve validation tests of the estimation model and the empirical strategy (pre-tests), estimation of the PSM-DID model, and validation of the results found with the estimations (post-estimation).

#### 2.3.1. Pre-Tests of the Model

The first pre-test verified the assumption of a “parallel trend” for the period before the implementation of the HGP, which also serves to validate the sample of counterfactuals selected for the DID model [30]. Although it is not possible to directly test the counterfactual hypothesis, the parallel trend was verified through the construction of a graph with the means of the outcome variable in the pre-treatment period (2007 to 2010) [35].

The Hausman test was used to test the hypothesis of endogeneity of the random term, and to verify the best functional form between the fixed effect and random effect models [29]. The third pre-test was the Wooldridge test, which aims to verify the presence of serial autocorrelation between the regression residuals [35,36]. Finally, the Wald test was performed to assess group heteroscedasticity in the regression residuals in panel models [37].

#### 2.3.2. Estimation of the PSM-DID Model

The implementation of the HGP took place through voluntary adherence, as its normative reference enabled municipalities to send in proposals for participation in the program from 2011 [15]. In this sense, although there were no restrictions on the submission of proposals, mayors of opposition parties to the federal government may have decided not to implement the program and have influenced the municipalities’ decision to implement the program or not. This non-random adherence could cause selection bias, due to the multidimensionality of factors and the local context, which may be related to the choice to implement the program [23,24,27,28]. In this sense, it was decided to use the PSM as a strategy to mitigate these problems and create groups of treated municipalities and controls that were statistically equal and, therefore, comparable in terms of their observable characteristics [23,27,38].

The matching procedure considered the entire study period (2007 to 2019) and was performed using logistic regression with a logit link function that considered the primary predictor variables and the variables that potentially influence the implementation of the HGP and expenses with hospital admissions for stroke. The matching was performed using the Kernel algorithm with 50 *bootstrap* repetitions. Kernel Matching is configured as an efficient metric to deal with units (municipalities) with different propensity scores, as it considers the weighted average of the control group to perform the matching and uses several or all untreated units as a control group for each treated unit [35,36,38].

The PSM generated the propensity scores, and the matching of municipalities generated the weights that were used to weight the estimates in the difference-in-difference model, configuring the PSM-DID method [39]. Then, the balancing test was carried out in order to verify statistical similarities between the matched variables before and after the implementation of the HGP, both at a level of 5%. Finally, the percentage of participation of municipalities in common support was calculated.

In this study, the difference-in-difference estimator explores the variations before and after the implementation of the HGP, with regard to hospital admission expenses for stroke between treated and control municipalities. It is worth noting that although the ordinance establishing the HGP was published in 2011, the adhesion of the municipalities occurred gradually, over the subsequent years. In this sense, the difference-in-difference model estimated in this study considers multiple implementation periods between 2011 and 2019 and fixed effects were added by municipality and by year.

The econometric formulation used in this study utilizes a binary variable (HGP) that simultaneously indicates whether the municipality was treated and in which year the implementation took place. For the control (untreated) municipalities, this variable assumes a value of zero for the entire time interval of the study. For the treated municipalities, the variable “HGP” assumed the value one in the year in which the municipality implemented the program and in subsequent years, and the value zero for the period prior to the implementation in that municipality.

Panel data models with fixed effects usually generate results more consistent with the cluster specification in Stata (“vce cluster” option), and are therefore preferable to the robust standard error specification “rob” [40,41]. In this sense, the standard errors of the PSM-DID estimations were calculated from a robust variance-covariance matrix per cluster of municipalities in order to correct eventual problems of serial autocorrelation of residuals and heteroscedasticity [42,43].

The results were reweighted by the residual variance of the units (in our case, municipalities), in order to minimize possible consequences of heteroscedasticity and improve the quality of the model’s fit. Thus, the variable code of the municipality was selected as the weight, considering that the variable to be weighted by the ‘weight’ must be constant within the units of analysis.

All monetary values were adjusted for inflation, based on the National Consumer Price Index (IPCA) accumulated between January 2007 and December 2019. The amounts were then converted from Real to US Dollar, using the exchange rate on December 31, 2019 (US$ 1.00 = R$ 4.03).

#### 2.3.3. Validation Post-Tests of the Results Found

The first test sought to identify the degree of correlation between the variables investigated and the treatment, and used a structure of *leads* and *lags* to verify the effects of pre-treatment and post-treatment (treated) [44,45]. The leads (anticipations) were inserted in the DID model to verify if the behavior of expenditures on hospitalizations for stroke after the implementation of the HGP already existed before the municipalities joined this intervention. The lags (delays) were inserted to verify if the effect of the program diminished after its implementation.

The second post-estimation test employed was the falsification test, also known as the placebo test. This test estimated the effect of the HGP on expenses with hospitalizations for stroke but with a placebo-dependent variable that, from a theoretical point of view, is not directly influenced by the effects of the program. The placebo variable chosen was the frequency of hospitalizations for arterial hypertension in the same period and the same municipalities.

## 3. Results

The results are presented in four sections. The first contains the descriptive statistics of the health, demographic, and socioeconomic variables of the treated and control municipalities. In the next section, the pre-test statistics are presented. The third section presents the estimation of the PSM-DID model used to measure the impact of HGP on expenses with hospitalizations for stroke. The fourth section presents the results of the post-estimation and robustness tests of the model.

### 3.1. Health, Demographic, and Socioeconomic Characteristics of the Municipalities

From 2007 to 2010 the spending with hospitalization for all causes was US$ 673,574,004.37 (annual mean = US$ 168,393,501.09; SD ± 105,577,808.02), including US$ 1,968,234.84 (annual mean = US$ 665,592.20; SD ± 408,252.16) for stroke (0.39% of all hospitalizations). In the period after the implementation of the HGP (2011 to 2019) the expenditure was US$ 3,013,037,284.81 (annual average = US$ 334,781,920.53; SD ± 49,432,360.61) with hospitalizations for all causes, including US$ 50,173,563,865 (annual mean = US$ 5,574,840.42; SD ± 2,435,505.96) for stroke admissions (3.03% of all admissions).

It should be noted that the treated municipalities had higher expenses with hospitalizations for stroke, and better health care indicators (number of doctors and hospital beds in the public network) than the control municipalities. In addition, health expenditure (transfer of federal financial resources to municipalities) in the treated group was more than twice that of the control group. Table 1 presents the health, demographic, and socioeconomic characteristics of the municipalities that implemented and did not implement centers of the Health Gym Program in the period from 2007 to 2019. The values were calculated from the means between the municipalities.

### 3.2. Model Estimation Pre-Tests

Figure 1 presents the trends in mean spending on hospital admissions for stroke in Pernambuco from 2007 to 2011 for treated and control municipalities and served to verify the assumption of parallel trajectories of the DID method. It can be observed that in the pre-treated period, the lines present slopes that indicate that the mean expenses of the dependent variable of the treated and control municipalities followed the same trajectory before the implementation of the HGP. However, after the year 2011, the slope of the expenditure line of the treated municipalities decreases, indicating that the expenditure on hospitalizations for stroke among the municipalities that implemented the HGP became lower than that of the comparison group.

The Hausman test result was statistically significant at the 1% level (Prob > chi2 = 0.0002), which indicates that the fixed effects model is more adequate to the data than the random effects model. The heteroscedasticity test (Wald test) also showed a statistically significant result at the 1% level (Prob > chi2 ≤ 0.001). In this sense, the null hypothesis that the model is not homoscedastic was rejected. Finally, the Wooldridge test indicated that there is no serial autocorrelation of the regression residuals (Prob > F = 0.0017).

### 3.3. Estimation of PSM, DID, and PSM-DID Models

All estimated models used the same variables, namely: the number of doctors, number of beds in public hospitals, population over 40 years of age, high school pass rate, GDP per capita, total health expenditure, and presence of Multiprofessional Support Teams for primary health care actions (NASF-AB). It should be noted that the propensity score pairing was performed both with the inclusion of the dummy variable for outlier and without it, but in the latter case, the ATT statistics showed a result with a positive and non-significant sign (ATT = 0.0617; T-stat = 0.44; S.E = 0.1524).

Table 2 presents the means of the variables for the treated and control municipalities before and after matching, and the balancing conditions in the treatment distribution. The result of the means comparison test enables us to state that after matching the groups became statistically equal (and different only in relation to the presence of HGP). It is noteworthy that, except for the variable presence of Multiprofessional Support Teams for primary health care actions (NASF-AB), all the other variables showed a bias reduction greater than 90% in the balance test. However, the Rubin’s B (19.0) and Rubin’s R (0.83) statistics demonstrate that the treated and control groups are sufficiently balanced.

The results for the effect of the HGP on hospital admission expenses for stroke are presented using the difference-in-difference and PSM-weighted difference-in-difference (PSM-DID) estimators. Table 3 presents the estimations performed. The results referring to the DID method served as a reference for the elaboration of the PSM-DID model, and are configured, in themselves, as a strategy for evaluating the impact of the HGP. However, the main result refers to the estimation through the PSM-DID method, which pointed that hospital admission expenses for stroke were 17.93% lower in municipalities that implemented the HGP (treated), when compared to municipalities that did not incorporate the intervention (controls). Regarding the PSM-DID model, the savings were 17.85%, and in both models the results were statistically significant at the 5% level.

### 3.4. Post-Estimation and Model Robustness Tests

The falsification test indicates that the treatment variable (presence of HGP) did not impact the placebo outcome (frequency of hospitalizations for arterial hypertension). Table 3 presents the coefficients of the DID, PSM-DID estimates and the placebo regression used as a robustness test.

The post-estimation test for leads and lags showed non-significant results at the 5% level for the treatment variable (presence of HGP in the municipality), which indicates that the model without anticipations or delays is adequate to measure the effect of the HGP on expenses with hospital admissions for stroke in the state of Pernambuco. However, both lag1 and lag2 were statistically significant, indicating the possibility that the effects may extend from one year to another. Table 4 presents the coefficients of the leads and lags test.

## 4. Discussion

In the current study, spending on hospitalizations for stroke increased between 2007 and 2019, reaffirming the growing trend in these expenditures across Brazil [46]. It is also noteworthy that the share of spending on hospitalizations for stroke as a part of spending on hospitalizations for all causes also increased. Barreto et al. [4] identified that in 2018, expenditure on hospitalizations for stroke represented 1.49% of expenditure on all hospitalizations, whereas in the current study, expenditure on stroke represented 3.03% of expenditure on all causes.

The municipalities that implemented the HGP presented higher expenses and greater availability of doctors and hospital beds in the public health network. This result corroborates the findings of Dantas et al. [46], who report that the non-qualification of hospital beds (especially ICU beds) by some municipalities burdens other cities that offer hospital services, making them responsible for funding the actions of urgency and emergency.

The first pre-test of the model verified whether the expenses with hospitalizations for stroke among the treated and control municipalities followed a parallel trajectory before the implementation of the HGP. This assumption of the DID method states that if the intervention did not exist, the time trajectory of the outcome variable should be parallel between the treated and control groups [45]. In this sense, it can be inferred that the unobservable characteristics interfere in the municipalities exposed and not exposed to the program in the same way (before implementation), indicating that the difference between the two groups may reflect only the mean effect of the program on expenditures with hospitalizations for stroke [47,48].

It is also worth noting that the drop in mean expenses before the implementation of the HGP and the increase observed in these values after its insertion in the municipalities corroborates the results of studies that point to a decrease in hospitalization expenses between the late 1970s and early 2000s and a subsequent increase in the frequency of stroke hospitalizations from 2009 [46,49].

The variables that made up the estimation models have already been pointed out in the literature as associated with greater hospitalization for stroke, reiterating the theoretical basis for the selection of the components of the model [30,31,32] and corroborating the results of studies that point to the influence of access to health actions and services, age, level of education of the population, and the Gross Domestic Product of the municipalities on the frequency of cerebrovascular diseases and on expenses with hospitalizations for stroke [7,30,31,50].

It is noteworthy that the presence of variables with coefficients that were not statistically significant (both in the PSM and in the PSM-DID) does not necessarily imply that they should not remain in the estimation models, as the removal of a variable can only occur in situations in which the evidence in the literature shows that it is not related to the outcome variable [38,51].

The test of difference in means before and after matching indicates that the hypothesis of equal means after matching cannot be rejected. In addition, the observable characteristics of the treated and control municipalities were satisfactorily balanced, given that Rubin’s B and R statistics (19.0 and 0.83, respectively) were within the limits established in the literature (B < 25 and 0.5 < R < 2, respectively) to test the balance quality [52].

Therefore, matching using the Kernel algorithm proved to be efficient to generate a control group similar to the treatment group, which enables the estimation of the impact of HGP on hospital admissions for stroke in Pernambuco.

The current study identified that the presence of the HGP reduces expenses with hospitalizations for stroke. Other studies have already evaluated the impact of the HGP in Pernambuco and found that the program reduced expenses with hospital admissions for cerebrovascular diseases [14] and mortality from systemic arterial hypertension [21].

Municipalities that implemented the HGP spent 17.85% less on hospital admissions than municipalities that did not adhere to the program. This impact could generate real savings of US$ 1,041,438.76 to the public health system over this period and represents 0.13% of the current revenue of the state health department in 2019 [53]. If the control municipalities joined the program, annual savings of approximately US$ 995,109.01 would be generated, which is equivalent to 0.075% of all expenditures by the state health department of Pernambuco in 2019 [54].

Considering that hospital care represents around 11.0% of the financial resources used in the treatment of patients who have suffered a stroke [8], the impact of the HGP in reducing public health expenditure may be even greater if we consider the potential savings with social security benefits, such as sick leave and disability pensions.

In a study carried out in the state of Pernambuco, Simões et al. [20] found that exposure to the presence of the HGP triples an individual’s chances of becoming physically active. In this sense, our findings may indicate that the effect of the HGP on the reduction in expenses with hospital admissions for stroke may be related to the potential increase in the level of physical activity promoted by the actions of the program, given that physical activity acts on risk factors for stroke, such as body weight and maintenance of blood pressure levels, as well as reducing the risk of stroke [12,55].

Another mechanism through which the HGP can contribute to the reduction in expenses with hospitalizations for stroke is the participation of the population in the health promotion activities that are developed by the program, especially those related to the adoption of healthy eating habits [15,16].

Evidence shows that costs with the implementation of the HGP between 2011 and 2017 were US$ 3,250,055,821.56 throughout the national territory [56]. In addition, considering that the program guidelines establish a monthly transfer of US$ 744.28 from the federal government to each program hub [15,56], it is estimated that US$ 198,724.78 was spent only for the Ministry of Health (not considering the municipal counterpart) with the implementation of the program only in the state of Pernambuco between 2011 and 2019. In this sense, it is recommended to carry out cost-to-effectiveness studies of the HGP, to verify if this intervention is cost-effective for stroke prevention and control in the state of Pernambuco and throughout Brazil.

Finally, with regard to the validity and robustness of the results found in this study, it is worth noting that the falsification test showed that the placebo variable (frequency of hospitalizations for arterial hypertension) was not impacted by the presence of the HGP in the municipalities, indicating that the results are being directed by the treated group [25]. In this sense, it is possible to affirm that the empirical strategy used in this study was adequate to assess the impact of the HGP on expenses with hospitalizations for stroke.

## 5. Conclusions

The Health Gym Program had an impact on the reduction in public spending on hospital admissions for stroke by 17.85% when comparing the municipalities that implemented this intervention with those that did not. It is noteworthy that the estimated savings in financial resources in this study represents only a small fraction of the expenditure on care for patients suffering from this disease, since it does not consider expenses with medication or eventual expenses with rehabilitation.

It should also be pointed out that although this is not the objective of this study, it is possible to infer that the presence of the HGP can also impact on the reduction in indirect costs with loss of productivity and social security expenses with temporary absences from work and early retirement.

The findings of this study can support decision-making processes on expanding the scope of the program in municipalities, as well as justifying public investment in its implementation or expansion.

### 5.1. Limitations and Future Studies

While our study contributes to the literature in using robust methodologies to evaluate the effect of the HGP on public health expenditures on hospital admissions for stroke, there are some limitations to note. First, data from only one state in Brazil may compromise the generalization of the results, both for Brazil as a whole and for other units of the federation. Meantime, it is important to highlight that, although caution is necessary when making inferences about this study, its findings have good internal validity to assess the effects of HGP on expenses with hospitalizations for stroke. The second limitation is due to the fact that this study did not use data at the individual level. On the other hand, evidence from studies that adopt similar methods but had used aggregated data showed results as robust as those with individual level data [23,24,27,28]. Third, the absence of variables that can interfere in the decision taken by the municipality to join the program stands out, such as the mayor’s political party at the time of implementation of the HGP. In this sense, the fact that the mayor belongs to a party allied to or opposed to the federal government can influence the choice of adopting the program or not. It should be noted, however, that this limitation was resolved both by the use of propensity score matching, which allows comparisons between the characteristics of municipalities exposed and not exposed [57] to the HGP, and by the use of the difference-in-difference estimator, whose properties minimize potential biases caused by the non-insertion of observable characteristics [23,24,25,26,27,28].

### 5.2. Implications to Public Health

The findings of the impact of HGP on expenses with hospitalizations reinforce the importance of public investments in health promotion policies designed to nudge changes in lifestyles. In addition, program managers and public policymakers will be able to use the evidence generated from this study to be accountable to the population, oversight agencies, and the legislature about the investment (and potential savings of resources) related to the implementation and adherence to the HGP.

The findings of this study reinforce the scientific literature that points out that policies aimed at improving the scope of actions in primary health care have great potential to prevent and control non-communicable chronic diseases [32,39,58], and save public resources, which can be reverted to other health actions and services. In this sense, this study helps managers and policymakers to report to the population, regulatory bodies, and legislature on investment (and potential savings in resources) related to the implementation of the HGP.

The results of this study may justify the opportunity cost related to the public investment that is carried out for the implementation of this program. Further, this study can generate evidence that support decision-making processes related to the expansion of the HGP, either with resources from the municipalities, the federal government, private companies or parliamentary amendments, as described in the program guidelines [17].

Finally, the findings of this study can help to evaluate the effectiveness of health promotion, preventing, and controlling actions in Primary Health Care, which are part of the Ministry of Health’s Strategic Action Plan for Coping with Chronic Noncommunicable Disease, mainly because the Health Gym Program is pointed out as the main strategy of health promotion in the Brazilian public health system [59].

## Figures and Tables

**Figure 1 ijerph-19-12174-f001:**
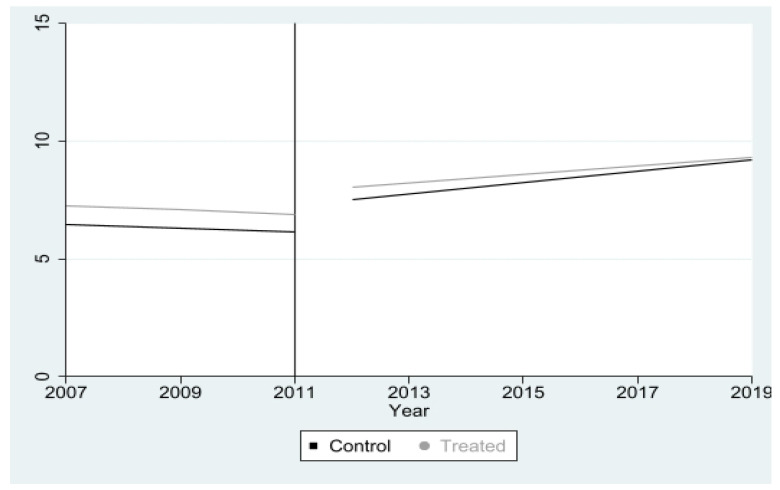
Trend in mean expenditure on hospitalizations for stroke in treated and control municipalities. Pernambuco, 2007 to 2019.

**Table 1 ijerph-19-12174-t001:** Health, demographic, and socioeconomic characteristics of the municipalities that implemented and did not implement centers of the Health Gym Program, Pernambuco, 2007 to 2019.

Variables	Control (0)		Treated (1)		Relative Difference in Mean * (0/1)	*p*-Value
	Mean *	SD	Mean *	SD		
Health						
Hosp spend per stroke **	4621.18	1155.20	11,886.13	1169.94	−7264.94	<0.001
No. of doctors **	14.22	0.92	68.94	9.3	−54.71	<0.001
No. of beds	38.32	2.01	115.51	13.07	−77.19	<0.001
Demographic						
Pop > 40 years ***	3847.14	119.76	8690.88	587.06	−4843.74	<0.001
Rt pass HS	86.27	0.33	86.72	0.2	−0.45	0.244
Socioeconomic						
GDP per capita	275,452.22	16,022.33	289,650.13	10,824.95	−14,197.90	0.480
Total Health Expenditure	4,848,757.54	151,985.75	11,150,605.58	904,148.96	−6,301,848.04	<0.001

***** The calculation of the mean took absolute values as a reference, but these variables underwent transformation (natural logarithm) to compose the models for evaluating the impact of the HGP; ** The calculation of the average took absolute values as a reference, but these variables were transformed (natural logarithm) to compose the PAS impact assessment models; *** The calculation of the mean took absolute values as a reference, but this variable underwent transformation (rate of people > 40 years old per 10,000 inhabitants) to compose the models for assessing the impact of the HGP; Source: produced by the authors. Note: *t*-test for difference of means. Legend: hosp: hospitalizations; No: number; Pop: population; Rt pass HS; High School pass rate; mi: million Reais.

**Table 2 ijerph-19-12174-t002:** Test of difference in means of treated and control groups before and after matching, balance test and common support of propensity score matching. Health Gym Program—2007 to 2019.

Variables	Before Matching			% Bias Reduction	After Matching		
	Treated	Control	*p*-Value		Treated	Control	*p*-Value
>40 years/10,000 inhab	3154	3005.4	<0.001	97.9	3138.7	3141.8	0.811
Log no. of doctors	2.52	2.064	<0.001	94.8	2.305	2.329	0.551
No. of hosp beds. SUS	116.56	39.452	0.001	95.8	53.416	50.187	0.218
Presence of NASF	0.586	0.561	0.292	−1.4	0.568	0.593	0.161
Total Health Expenditure	105,912,295.91	44,314,883.50	<0.001	99.1	55,848,149.88	55,507,149.30	0.780
Rt pass HS	86.138	85.631	0.211	92.6	86.286	86.324	0.902
GDP per capita	2,753,400.11	2,816,644.81	0.481	90.7	2,731,272.41	2,727,414.55	0.923
**Balancing Conditions (Rubin statistics)**							
B	19.0						
R	0.83						
Panel B—Common support of matching between treated and untreated groups							
Out of Support		Common Support		Total		% of Participation	
Control	0	606		606		100	
Treated	92	1509		1601		94.25	
Total	92	2115		2207		95.83	

Source: produced by the authors.

**Table 3 ijerph-19-12174-t003:** Impact of the Health Gym Program on hospital admissions for stroke, and placebo regression coefficients. Pernambuco, 2007 to 2019.

Variables	DID		PSM-DID		Placebo Regression	
	Log Stroke Expenditure *	Standard Error	Stroke Expenditure *	Standard Error	Hosp for Hypertension	Standard Error
HGP	−0.1793 ^b^	0.089	−0.1785 ^b^	0.089	0.1447	−2.302
Propensity Score	-	-	1.228	−1.051	−7.631	30.65
>40 years/10,000 inhab	0.002 ^a^	<0.001	0.002 ^a^	0.001	−0.011	0.019
Log no. of doctors	−0.093	0.065	−0.118 ^c^	0.067	−0.442	0.993
No. of hosp. Beds.	−0.000	0.001	−0.001	0.001	0.005	0.017
Presence of NASF-AB	−0.014	0.086	0.088	0.111	−2.784	−3.115
Total Health Expenditure	0.000 ^a^	<0.001	<0.001 ^a^	0.000	−0.000 ^b^	<0.001
Rt pass high school	0.014 ^a^	0.005	0.014 ^a^	0.005	−0.024	0.122
GBP per capita	0.000 ^a^	<0.001	<0.001 ^a^	<0.001	<0.001	<0.001
outlier	−7.208 ^a^	0.140	−7.176 ^a^	0.145	0.760	−1.890
Time of Exposure						
1st Year	−0.436 ^c^	0.251	−0.346	0.258	−1.780	−8.426
2nd Year	−0.487 ^a^	0.233	−0.408 ^c^	0.237	−1.328	−7.388
3rd Year	0.108	0.195	0.174	0.201	−1.324	−6.066
4th Year	0.366 ^b^	0.168	0.416 ^b^	0.173	−0.695	−4.713
5th Year	0.222	0.148	0.268 ^c^	0.153	−2.372	−3.842
6th Year	−0.034	0.110	0.003	0.116	−1.719	−2.491
7th Year	0.092	0.0913	0.121	0.097	−0.874	−1.588
8th Year	−0.452	0.411	−0.581	0.421	0.140	11.155
Constant	0.962	1.842	0.962	1.842	65.03	56.18
R^2^	0.855		0.855		0.134	

^a^*p* < 0.01, ^b^ *p* < 0.05, ^c^ *p* < 0.1. * Natural logarithm of expenditure on hospital admissions for stroke. Note: Robust standard errors clustered at the municipality level. The exposure time starts in 2011. Legend: log = natural logarithm; inhab = inhabitants; hosp = public hospitals; Rt pass high school = high school pass rate.

**Table 4 ijerph-19-12174-t004:** Leads and Lags test.

Stroke	Coeficiente	Standard-Error	z	*p*-Value	95% Confidence Interval	
lead2	0.252	0.197	10.28	0.201	−0.134	0.64
lead1	0.081	0.193	0.42	0.675	−0.298	0.461
treat	−0.023	0.191	−0.12	0.904	−0.398	0.352
lag1	−0.45	0.191	−20.35	0.019	−0.825	−0.075
lag2	−0.596	0.191	−30.12	0.002	−0.971	−0.222
_cons	7.95	0.128	61.86	<0.001	7.703	8.207

The series comprises 185 municipalities, over a period of 13 years (2007–2019), totaling 2207 observations. Treated variable (presence of HGP). Leads stats for pre-trend, lead1 = 1 year lag, lead2 = 2 years lag. Lag statistics for post-trend, lag1 = 1 year lag, lag2 = 2 years lag. Source: Research data. Legend: Stroke = expenditure on hospital admissions for stroke.

## Data Availability

Some or all data and models that support the findings of this study are available from the corresponding author upon reasonable request.

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
