# Peer review of "Health Promotion Programs Can Mitigate Public Spending on Hospitalizations for Stroke: An Econometric Analysis of the Health Gym Program in the State of Pernambuco, Brazil"

_ijerph, 2022, doi:10.3390/ijerph191912174_

Round 1

Reviewer 1 Report

The study entitled "Health promotion programs can mitigate public spending on hospitalizations for stroke: an econometric analysis of the Health Gym Program in the state Pernambuco, Brazil", presents an excellent theoretical description, is well written and addresses a relevant topic for Brazilian public health .

The study has a clear objective and its results, as well as the conclusion of the study, respond well to the proposed objective.

Small corrections are needed:

Summary:

Acronyms appear in the abstract, but their description is not given. I suggest writing in full in the abstract.

Introduction:

The introduction is well written, follows a line of reasoning and brings relevant information. However, the authors do not make it clear what is the real gap in the literature that they intend to fill.

Author Response

Dear Reviewer,

Initially, I would like to thank you for your careful reading of the text and for your valuable contributions to qualifying my manuscript.

Below I present the responses to your valuable comments

1 - Recommendation to remove acronyms from the abstract -  suggestion accepted. I inserted the name of the program instead of the acronym.

2 – gap is not clearly described in the introduction - Suggestion was accepted. The information about the GAP was presented more clearly in the last paragraph of the introduction.

Reviewer 2 Report

Generally, this paper is rather interesting, it evaluates the effects of the Health Gym Program (HGP) on expenditures on hospitalizations for stroke in the state of Pernambuco, Brazil. There are some areas that are not clear to me and where I believe the author could improve.

1.    I do not understand why you consider the entire period 2007-2019 to perform the PSM. Your analysis would be more robust if you consider only data collected before the intervention.

2.    Table 1 may be more informative if you report the statistics before and after the intervention, for both Control and Treated.

3.    Line 132, as a robustness check you could drop all the outliers

4.    The first paragraph of section 3.1 could be made easier to read.

Minor comments:

1.    Line 43, I do not understand why you refer to the state of Minas Gerais

1.    Line 112, the independent variable it is only the exposure to HGP, right? All the other variables are control variables.

2.    Line 137, don’t need to write which software you used in the main text.

Author Response

Dear Reviewer,

Initially, I would like to thank you for your careful reading of the text and for your valuable contributions to qualifying my manuscript.

Below I present the responses to your valuable comments

Questioning about doing the matching using the entire study period - In fact, matching the whole study period could create situations in which a municipality treated at t+1 (post-treatment period) is paired with a municipality treated in the time t (treatment period), which would not be adequate.

The best scenario would be to match a municipality treated at t (treatment period) to a municipality at t–1 (pre-treatment period). However, in the specific case of this study, the municipalities joined the HGP at different times in the period between 2011 and 2018. In this sense, it may be that some characteristics of the municipalities are dynamic and that in the following years the similarities between these two municipalities change, which would make pairing in the period prior to treatment inefficient.

However, according to Lechner (1999), Nielsen and Sheffield (2009), current matching algorithms match treated only with untreated ones. In this case the match using the entire period of study can avoid situations in which, for example, a municipality treated at t +1 (post-treatment period) be matched to another treated municipality, i.e., current matching methods pose no risk of matching treaty to treated. In addition, they minimize problems related to eventual changes in the characteristics of observation units over time.

References:

NIELSEN, R.; SHEFELD, J. Matching with time-series cross-sectional data. Polmeth XXVI. Yale University, 2009.

LECHNER, M. Earning and Employment Effects of Contínuos Off-the-job Training in East Germany After Unification. Journal of Business & Economic Statistic, Taylor & Francis Group, v.17, no. 1, p. 74-90, 1999.

Suggestions about change table 1 - The table with the values of the descriptive statistics before and after the implementation of the program could be more informative, but as this table presented, it is possible to verify important differences between the treated and control municipalities, even without the detailing of values before and after the HGP deployment. In addition, changing the table would generate results very similar to those presented in table 2, and would require important changes in the discussions. In this sense, I very respectfully request that you consider maintaining the table in the format that it is, considering that the other reviewers did not request changes to the table in question and that any changes may imply altering both the requested corrections and the assessment that the other reviewers have already made of the article.

Comment regarding the removal of outliers - In fact, the removal of outliers is a strategy to increase the robustness of the results however, we decided to use the outlier dummy as a strategy to verify the weight of these extreme values in the estimation of the impact, and therefore the we kept it in the models presented in table 2.

According to Brooks (2019): “…The effect of the dummy variable is exactly the same as if we had removed the observation from the sample altogether and estimated the regression on the remainder”. It is important to highlight that this technique is used in studies that use econometric analyses, such as Barata (2016), Keenan (2020).

References

Brooks, C. (2019). Introductory Econometrics for Finance (4th ed.). Cambridge: Cambridge University Press. doi:10.1017/9781108524872.

João Barata Ribeiro Blanco Barroso, 2016. "Quantitative Easing and United States Investor Portfolio Rebalancing Towards Foreign Assets," Working Papers Series 420, Central Bank of Brazil, Research Department.

Keenan, Alison, "Understanding the Temporary Assistance for Needy Families' Reach Across the U.S." (2020). Honors Theses. 83

Johansen, S. and Nielsen, B. (2016). Asymptotic Theory of Outlier Detection Algorithms for Linear Time Series Regression Models. Scandinavian Journal of Statistics, 43(2):321–348.

The first paragraph of section 3.1 could be made easier to read - The recommendation was accepted and the text was rewritten.

Questioning about the reference to the state of Minas Gerais in line 43 - The text was rewritten, to better contextualize the information.

Suggestion to replace the term “independent variables” with control variables in line 112 - Suggestion accepted. The correction was performed.

The recommendation that it is not necessary to quote the software name in the text - the option to quote the software name was given because in some parts of the text I mention specific commands and routines used in data analysis as occurs for example in line 196. In this regard, I very respectfully request that you consider keeping the name of the statistical package.

Reviewer 3 Report

This is a rigorous, in-depth analysis of a novel health promotion program’s effect on hospitalization costs and is a good model for policy interventions to improve health. The author puts forth a strong analysis; more insight with respect to details of the Health Gym Program, why some municipalities implemented the program and others did not would further strengthen the manuscript. and potential selection

Some details are missing about why some municipalities implemented the Health Gym Program and why other municipalities did not implement the program.  Do you have any insight regarding how these decisions were made?

Concerns with Analytic Approach

There are more sophisticated methods for analyzing data with 0 costs. Adding one unit is a relatively crude approach. Please consider other methods.

I am not familiar with the approach of including a binary variable to indicate whether a municipality is an outlier in the analysis. Please provide references and/or sensitivity analyses to including versus excluding these observations in the analysis. Along the same lines, trimming the data with respect to propensity scores is another approach to addresses outliers. Please describe how you addressed outliers with respect to the propensity score versus outcomes more generally.

Given the similarity of findings between the DID and PSM-DID approaches, what is the value of using the PSM-DID? If there is value in the PSM-DID model, is it necessary to report both in the main tables? It is not clear that the two sets of results add value.

Discussion

The discussion spends a great deal of time describing the analytic approach, however, these paragraphs need more translation to policy or practice. Along the same lines, what is the potential cost savings annually if the control groups implemented the Health Gym Program? Additionally, the econometric analysis does not take into account the cost of implementing the Health Gym Program. Insight regarding the implementation cost versus cost savings of the program would strengthen the discussion.

Other Concerns

Please describe the data source or sources and the validity of these data sources.

Figure 1 – are these actual mean values for each year, or a trend line? The actual mean values would be more informative than a trend line.

Minor comments

Line 54: This paragraph discusses the cost savings of a health promotion program, stating that municipalities implementing the intervention spent $1259 per 10000 inhabitants compared to municipalities that did not implement the intervention. Is this is a large or small amount, in context of community-level interventions? It is difficult to put this dollar amount into context.

P-values are <.001 rather than .000

Table 2 – it is unclear what “Presence of NASF” is. This variable is not described in the text.

Author Response

Dear Reviewer,

Initially, I would like to thank you for your careful reading of the text and for your valuable contributions to qualifying my manuscript.

Below I present the responses to your valuable comments

1 – Why do some municipalities incorporate, and others do not

As it is a voluntary adhesion to a strategy financed by the federal government, the mayors of opposition parties may have decided not to implement the program however, there is no evidence of this. I have inserted some sentences in the first paragraph of item 2.2.3 to make this information clearer to the reader.

2 – Strategy of adding a unit to deal with observations with zero value -->  Certainly the recommendation to use other methods is precious and would greatly enrich the work under analysis. However, this would require redoing all the pre-analyses, the main modeling, the tables, and the post-estimation tests. In this sense, considering that the strategy used, although unsophisticated, is recommended in classic econometric manuals, such as the book by Professor Jeff Wooldridge (2016) and that the other reviewers of the article did not request such a correction, I very respectfully request to consider the strategy used for this paper and gratefully welcome the recommendation to adopt other methods in future publications.

3 – Use of the outlier variable --> I provided the reference and some clarifications regarding the strategy. In addition, I pointed out some econometric studies that used the outlier variable. According to Brooks (2019): "…The effect of the dummy variable is exactly the same as if we had removed the observation from the sample altogether and estimated the regression on the remainder”. It is important to highlight that this technique is used in studies that use econometric analyses, such as Barata (2016), Keenan (2020).

References

Brooks, C. (2019). Introductory Econometrics for Finance (4th ed.). Cambridge: Cambridge University Press. doi:10.1017/9781108524872.

João Barata Ribeiro Blanco Barroso, 2016. "Quantitative Easing and United States Investor Portfolio Rebalancing Towards Foreign Assets," Working Papers Series 420, Central Bank of Brazil, Research Department.

Keenan, Alison, "Understanding the Temporary Assistance for Needy Families' Reach Across the U.S." (2020). Honors Theses. 83

Johansen, S. and Nielsen, B. (2016). Asymptotic Theory of Outlier Detection Algorithms for Linear Time Series Regression Models. Scandinavian Journal of Statistics, 43(2):321–348.

4 – Questioning how the results of the outliers were used in the propensity score --> I added a sentence explaining that PSM models were generated with and without the outlier variable (comparison of the general results versus the results with the outlier variable), and I inserted the statistics that justify the choice to include the outlier variable in the PSM model (page 7, item 3.3).

5 - Questioning about the importance of presenting the results of the DID and PSM-DID estimations in table 2 -->  I added information indicating that the results of the DID estimation were used only for the elaboration of the PSM-DID model and that the latter is configured as the main result of the study.

6 – Suggestion to broaden the discussion on the application of results in the field of public policies --> The suggestion was accepted and I inserted three more paragraphs on the implications for public health on page 12.

  7 – Question about what is the potential cost savings annually if the control groups implemented the Health Gym Program --> Suggestion accepted. The information was added on page 10.

8 – Suggestion to add a discussion about implementation cost versus cost savings of the program would strengthen the discussion --> Suggestion accepted. I added a paragraph to page 11 of the article.

9 – Recommendation to describe data sources --> Recommendation accepted. I included a paragraph on data sources on page 3.

10 – Suggestion to remove figure 1 --> Figure 1 aims to graphically represent the parallel trend of the dependent variable and the change in the trajectory of these averages between treated and controls in the years after implementation. The suggestion of replacing the figure for the average values is pertinent and valuable. However, I believe that the illustration can contribute not only to facilitating the visualization of the parallel trend before the implementation of the program (which is one of the assumptions of the DID method), but also to representing the change in the behavior of the average expenditure, which had a downward trend in the first years of the series but increased from 2009 onwards. In this sense, I very respectfully suggest that the figure be mantained in the text, as it serves as a support for the discussion of the results regarding the increase in expenses with hospitalizations for stroke in the state, which is presented on page 10 (Refs 43 and 46).

11 – Suggestion to contextualize a value expressed in the text of line 54 --> I added the information on page 2.

 12 – Questioning about the variable “Presence of NASF” in table 2 --> The variable indicates the presence of Multiprofessional Support Teams for primary health care actions, and is described in line 118.

13 – Suggestion for formatting p-values in tables --> Suggestion accepted. I corrected the tables, both for p-values and standard errors.
